# OpenReview forum: "Automated Red Teaming with GOAT: the Generative Offensive Agent Tester"
_ICLR.cc/2025/Conference — Submitted to ICLR 2025_

### Official Review · Reviewer_7aNr · 2024-11-01

**Soundness:** 2
**Presentation:** 3
**Contribution:** 2
**Rating:** 5
**Confidence:** 4

**Summary:**

This paper introduces GOAT, an agent framework to automatically jailbreak models with multi-turn attacks. The paper evaluates GOAT on JailBreakBench and demonstrates its ability to jailbreak several frontier closed-source and open-source models. GOAT outperforms another multi-turn automated jailbreaking method, Crescendo.

**Strengths:**

- The paper provides an automated agent to jailbreak the most powerful systems. Additionally, the type of attacks the system generates are more realistic and therefore close in distribution to the attacks from real users. Hence this agent is more valuable for auditing than other jailbreaking techniques.
- The agent achieves SOTA and is automated, suggesting that this is a reasonable auditing technique for current models.
- The agent should scale as more jailbreaks get discovered.
- The paper is well-written and the qualitative examples + figures are helpful for understanding.

In summary, the paper is well-executed (good quality), well-written (good clarity), and tackles the challenge of identifying multi-turn jailbreaks (reasonable significance).

**Weaknesses:**

While this paper is useful, I think it misses an important opportunity to help develop community norms or guide research on multi-turn jailbreaks. This miss is amplified by the fact that this paper appears one of the first automated multi-turn jailbreaking methods. Concretely, the paper:
- **Lacks conceptual insights.** I gained almost no additional information from reading the paper compared to reading only the abstract + Figure 1. Specifically, I recommend
     -  Explore the design space of multi-turn jailbreaking agents by ablating different components of the agent. While the authors do not need to incorporate all of following suggestions, I think exploring a few of these design decisions could make the paper much more informative for future work. I've italicized the top three imo:
         - *How much does the reasoning part of the prompt help? Can you ablate this part of the prompt?*
         - *Does using a stronger LLM improve ASR?*
         - What is the impact of training data on the attacker LLM?
         - Is the attacker LLM still able to succeed with fewer jailbreaks?
         - How does the ASR scale with number of jailbreakers?
         - *Are there some jailbreaks that are more important s.t. removing them dramatically reduces performance?*
         - Does passing evaluation of success to an external judge help (at the expense of more API calls)?
         - Is it possible to have the attacker LLM be a heavily safety-trained model?
         - Does increasing the diversity of initial strategies tried by the LLM improve the ASR?
    - Explore the realism of the data generated by GOAT. Understanding the following questions would help understand the qualitative nature of the method more precisely:
         - Given the human-written distribution of mutli-turn jailbreaks in [1], how similar are the jailbreaks generated by GOAT?
         - Could training to refuse the jailbreaks from GOAT serve as a defense method against human jailbreaks?
         - Is it possible to make GOAT attacks match the distribution of human attacks?
- **Has weak empirical rigor.** I think some of the experiments / writing could be clarified, and I hope the authors will address this. In particular,
    - What exactly is the attacker LLM? How was it trained? The paragraph from L296-304 is extremely vague.
    - Does GOAT work with other attacker LLMs? Can I just feed the attacker prompt directly into GPT-4o or Claude-3.5-Sonnet and have it act as my attacker LLM?
   - GOAT appears to be plateauing at a faster rate than Crescendo. Will the two ASRs @ 10 match with enough turns? What does the graph looks like with 10 turns of dialog?
- **Lacks originality.** This is mostly tied to the first weakness, so I think this is more of a minor issue. Given that there aren't many scientific insights in the paper, I feel like the paper itself isn't very novel. In particular, it feels like a multi-turn agent with CoT prompting applied to jailbreaks. Although I think prompting is very valuable, there doesn't seem to be any particularly new ideas in this paper. It would be cool if the authors could consider implementing an additional 'feedback stage' to see if the agent could generate its own jailbreaks given examples of previously failed trajectories / strategies.

I'm willing to adjust my score to a 6 if some of the weaknesses are addressed, especially the lack of empirical rigor.

[1] Li, Nathaniel, et al. "Llm defenses are not robust to multi-turn human jailbreaks yet." arXiv preprint arXiv:2408.15221 (2024).

**Questions:**

Suggestions:
- Lines L89 - 103 are verbose and don't seem to have any connection. Are they simply motivating that multi-turn jailbreaks should be dynamic? I feel like it would be helpful to shorten the exposition in the intro and instead focus on the insights gained from developing GOAT.
- The text in Figure 1 is small and too hard to read. It would be nicer if you shortened some of the text and had bigger blocks that highlighted the overall strategy of the agent.
- In L195, say where the 7 attacks were sourced from or how they were developed.
- Replace the algorithm with a table that highlights the different parts of the attacker prompt.
- Include more qualitative scenarios. I thought the red / blue tables were very helpful but there were only two included.

---

### Official Review · Reviewer_DUB6 · 2024-11-03

**Soundness:** 3
**Presentation:** 3
**Contribution:** 3
**Rating:** 6
**Confidence:** 3

**Summary:**

This paper presents GOAT, an automated agent-based red teaming framework designed for testing the robustness of large language models (LLMs) by simulating adversarial multi-turn conversations. GOAT leverages a general-purpose LLM as an attacker that utilizes multiple pre-defined adversarial prompting strategies to elicit policy-violating responses from target models. The methodology incorporates a step-by-step reasoning structure inspired by chain-of-thought prompting, allowing GOAT to adapt dynamically during conversations. The framework was evaluated against state-of-the-art models like GPT-4-Turbo and Llama 3.1, demonstrating a significant attack success rate (ASR@10) and outperforming existing multi-turn approaches such as Crescendo.

**Strengths:**

1. The framework achieved impressive results, with ASR@10 rates of 97% for Llama 3.1 and 88% for GPT-4-Turbo on the JailbreakBench dataset, surpassing existing methods.

2. GOAT is extensible, incorporating seven distinct adversarial strategies, such as Persona Modification and Response Priming, to simulate a variety of red teaming approaches.

**Weaknesses:**

1. The paper didn't mention which specific large language model (LLM) was used as the attacker in the GOAT framework.

2. The maximum number of conversation turns was capped at five due to context window limitations. This constraint might impact the effectiveness of certain attack strategies and limits the scope of adversarial escalation in longer dialogues.

**Questions:**

1. How does GOAT handle longer multi-turn conversations beyond the current limit of five turns, and what modifications would be needed to extend this capability?

2. What criteria were used to select the attacker LLM, and how might changes to this choice impact GOAT’s performance?

3. How adaptable is GOAT to LLMs that are continuously updated with new safety measures? Would it require significant re-training or additional adversarial strategy development?

---

### Official Review · Reviewer_jc6b · 2024-11-04

**Soundness:** 3
**Presentation:** 3
**Contribution:** 3
**Rating:** 6
**Confidence:** 4

**Summary:**

This paper proposes the Generative Offensive Agent Tester (GOAT), an automated red-teaming system that conducts multi-turn jailbreaking. GOAT allows the attacker to dynamically adapt its strategy based on the conversation context. It includes an action set with seven distinct attack strategies and is easily extendable. The paper demonstrates GOAT's efficacy on the JailbreakBench dataset, outperforming the Crescendo method. Notably, GOAT achieves a high attack success rate within just five conversation turns.

**Strengths:**

- The motivation for simulating human red-teaming is well-justified.
- The method offers strong flexibility due to its extendable set of attack strategies.
- This approach leverages the strengths of existing attack methods (seven prior strategies) along with the flexibility of multi-turn conversation.
- It achieves a high ASR within 5 turns, which is impressive.

**Weaknesses:**

- The method does not introduce particularly novel insights; the concept of multi-turn adversarial interaction has been explored in previous works.
- The paper primarily compares GOAT to only one existing method, Crescendo, without assessing its performance against other red-teaming or adversarial attack approaches.
- The paper mentions limitations of the attacker model’s capabilities, such as context window length and prompt-following strength, but it lacks empirical analysis of how these factors impact GOAT’s performance.

**Questions:**

- The paper describes the Attacker LLM as a "general-purpose helpful language model." However, the exact model used is unspecified. Key questions: What is the model type? What are the specific implementation details? How might the choice of model, particularly its capabilities and prompt-following strength, impact the results?
- Could you elaborate on the rationale for selecting smaller Llama models and larger GPT models as target models? How might GOAT perform on other popular models with varying sizes or levels of safety alignment? (Note: It is reasonable to use those models since we are always facing the limited resources.)
- Is GOAT’s multi-round approach strictly superior to other methods? How would it perform compared to alternative attack types, or other single-round and multi-round adversarial strategies?
- Given that the Crescendo attack requires more conversation turns to reach its peak performance, would GOAT still maintain an advantage if the number of turns were increased? How does this compare to other methods, and are there trade-offs associated with the number of queries?

Update: I have raised my score since many of my concerns have been addressed.

**Details Of Ethics Concerns:**

There is potential for misuse of this method, and it may be challenging to mitigate these risks.

---

### Official Review · Reviewer_k3uS · 2024-11-04

**Soundness:** 3
**Presentation:** 2
**Contribution:** 3
**Rating:** 5
**Confidence:** 4

**Summary:**

This paper introduces GOAT (Generative Offensive Agent Tester), an automated system for red teaming language models through multi-turn adversarial conversations. Unlike previous approaches that focus on finding single adversarial prompts, GOAT simulates how real users interact with LLMs through extended conversations. The system uses a general-purpose LLM as an attacker that can dynamically employ multiple jailbreaking techniques while maintaining a conversation. The authors demonstrate GOAT achieves high attack success rates with only 5 queries.

**Strengths:**

- Novel approach that addresses a realistic threat model - focusing on multi-turn conversational attacks rather than single-prompt attacks
- Well-engineered system that effectively combines multiple existing attack techniques in a dynamic way
- Strong empirical results showing high success rates with relatively few queries
- Extensible framework that can easily incorporate new attack techniques

**Weaknesses:**

- Evaluation only considers two model families (llama and gpt) and one baseline. Claude should be tested, as it is the most adversarially robust model. While I understand the lack of multi-turn baselines, it is possible to convert single-turn baselines such as PAIR to multi-turn by applying the attack multiple times.
- Limited analysis of why certain attack combinations work better than others - more insight into patterns of successful attack sequences would be valuable
- No evaluation or discussion of defenses [1,2]. To my knowledge all defenses were proposed for single-turn settings, would they still be effective in this setting?

[1] Robey, A., Wong, E., Hassani, H., & Pappas, G.J. (2023). SmoothLLM: Defending Large Language Models Against Jailbreaking Attacks. ArXiv, abs/2310.03684.
[2] Zhou, A., Li, B., & Wang, H. (2024). Robust Prompt Optimization for Defending Language Models Against Jailbreaking Attacks. ArXiv, abs/2401.17263.

**Questions:**

What makes the LLM select certain attacks over other ones, especially on the first iteration? Some analysis/discussion on this would be helpful

How does this automated attack compared to human multi-turn jailbreaks [1]? This is a major motivation in the paper but GOAT is not compared against human attacks. How do the attack strategies differ than how humans conduct multi-turn jailbreaks?

[1] Li, N., Han, Z., Steneker, I., Primack, W., Goodside, R., Zhang, H., Wang, Z., Menghini, C., & Yue, S. (2024). LLM Defenses Are Not Robust to Multi-Turn Human Jailbreaks Yet. ArXiv, abs/2408.15221.

---

### Official Review · Reviewer_Vo9c · 2024-11-04

**Soundness:** 3
**Presentation:** 3
**Contribution:** 3
**Rating:** 8
**Confidence:** 4

**Summary:**

This work offers a novel way to simulate and automate how human users may use publicly available jailbreak methods to circumvent the safety training of LMs in long multi-turn conversations. The proposed algorithm (GOAT) is successful and competitive for five tested models, cost-effective ro tun (e.g., compared to fine-tuning or gradient-based attacks), and easily updated as it relies on in-context examples.

**Strengths:**

- The proposed automated red-teaming algorithm is novel and elegantly simple, making it scalable and usable for future works
- The shown results demonstrate the potency of the attack for all five tested models
- The methodology is clearly communicated and the paper is generally well-presented and easy to follow
- Compared to other, more involved attacks, the proposed algorithm is cost-effective and can be expanded with newer jailbreak prompts

**Weaknesses:**

- The proposed algorithm is limited to "conversational" jailbreak prompts. While it offers a significant degree of automation and value to current model red-teaming, this attack approach might be easier defended against with more advances in safety training. Also, it is unclear if the proposed algorithm can be used to discover novel attacks. For example, would it work to ask the attack model to evaluate previous conversations and come up with a novel attack?

- The authors state that the paper's goal is not to compare models comprehensively; they only want to focus on a prominent model family for open-source and closed models. However, I think the results of this work could be significantly strengthened by adding more capable models in the chosen families (GPT-4o, or larger than 8B Llamas) and, ideally, adding a third model family like Claude or Gemini.

- The described methodology/experimental setup is vague on what LM was used for the attacker model. It is only described as a "generic helpful-only language model" and that it was "exposed" to numerous samples of being overcompliant instead of replying safely. The vagueness used to describe the model limits reproducibility. The contribution of this work to the ICLR community would be improved by either clearly stating how this model was created (base model, number of parameters, training data, fine-tuning data, ...) or at least stating some information that can be used to estimate the model capability (number of parameters and benchmark performance) necessary for the proposed algorithm to work and acknowledging that it is a proprietary model.

- The line widths, axis labels, axis ticks, plot legend, and plot titles are too small and unreadable in Figure 3. In addition, the font in Figure 1 is also small and hard to read.

**Questions:**

- Please address the concerns I see in the weaknesses. I am willing to increase my score if you extend your analysis to more models or demonstrate generalized usability in another way (e.g., discovering new attacks etc.)

- Please add a trigger warning for offensive content to your abstract! The presented harmful examples in the main body and appendix of a racist text and a nazi-glorifying poem require such a warning.

---

### Meta-Review · Area_Chair_1iEg · 2024-12-20

**Metareview:**

The reviewers for this paper have diverse assessments, with overall ratings for the paper being borderline. On the positive side, the reviewers acknowledged that the proposed automated red-teaming method leverages an interesting idea of multi-turn conservations simulating human red-teaming, and the method is effective and practically useful. On the negative side, the reviewers also raised several concerns about limited empirical analysis regarding considered baselines or attack combinations, lack of discussion/evaluation about potential defenses, and lack of systematic ablations to get better insights about multi-turn jailbreaking. We want to thank the authors for their responses and active engagement during the discussion phase. These responses did help in improving the reviewers' assessment of the paper; however, the paper's ratings still stand borderline. Nevertheless, this is potentially impactful work, and we encourage the authors to incorporate the reviewers' feedback when preparing a future paper revision.

**Additional Comments On Reviewer Discussion:**

This is a borderline paper leaning toward rejection. Only one reviewer has a high positive rating (8), championing the acceptance; the four other reviewers have borderline ratings (5,5,6,6). The reviewers raised several concerns about limited empirical analysis regarding considered baselines or attack combinations, lack of discussion/evaluation about potential defenses, and lack of systematic ablations to get better insights about multi-turn jailbreaking. The authors' responses did help improve the reviewers' assessment and ratings of the paper; however, some of the reviewers' concerns still remain unaddressed, and the paper is still borderline after discussions.

---

### Decision · Program_Chairs · 2025-01-22

Reject